# STRUCTURE-ENHANCED PROTEIN INSTRUCTION TUNING: TOWARDS GENERAL-PURPOSE PROTEIN UNDERSTANDING

## ABSTRACT

Proteins, as essential biomolecules, play a central role in biological processes, including metabolic reactions and DNA replication. Accurate prediction of their properties and functions is crucial in biological applications. Recent development of protein language models (pLMs) with supervised fine tuning provides a promising solution to this problem. However, the fine-tuned model is tailored for particular downstream prediction task, and achieving general-purpose protein understanding remains a challenge. In this paper, we introduce Structure-Enhanced Protein Instruction Tuning (SEPIT) framework to bridge this gap. Our approach integrates a noval structure-aware module into pLMs to inform them with structural knowledge, and then connects these enhanced pLMs to large language models (LLMs) to generate understanding of proteins. In this framework, we propose a novel two-stage instruction tuning pipeline that first establishes a basic understanding of proteins through caption-based instructions and then refines this understanding using a mixture of experts (MoEs) to learn more complex properties and functional information with the same amount of activated parameters. Moreover, we construct the largest and most comprehensive protein instruction dataset to date, which allows us to train and evaluate the general-purpose protein understanding model. Extensive experimental results on open-ended generation and closed-set answer tasks demonstrate the superior performance of SEPIT over both closed-source general LLMs and open-source LLMs trained with protein knowledge.

## 1 INTRODUCTION

Proteins are large biomolecules and macromolecules, composed of one or more long chains of amino acid residues, playing pivotal roles in catalyzing metabolic reactions, DNA replication, and other significant biological processes (Anfinsen & Haber, 1961; Hartley, 1951). Generally, proteins are represented in two types of forms: *one-dimensional (1D) sequence* detailing the order of amino acids, and *three-dimensional (3D) structure* illustrating the spatial configuration of the protein. The 1D sequence of protein is generated by transcribing and translating a gene's DNA sequence and folds into a specific 3D structure, and the 3D shape determines the protein's properties and functions (Pei et al., 2024b). Conventional machine learning methods (Radivojac et al., 2013; Whisstock & Lesk, 2003) have achieved notable accuracy in protein property and function prediction via supervised learning. However, these methods are all task-specific as each model is restricted to predicting a particular property. The increasing need for comprehensive protein analysis in various fields, such as pathology and drug discovery (Guo et al., 2023), calls for the development of general-purpose protein understanding models capable of accurately predicting various protein properties and functions.

In the fast-evolving era of large language models (LLMs), large efforts have attempted to leverage their capabilities of semantic understanding and complex reasoning to achieve general-purpose protein property and function prediction. Initially, some methods treat the 1D protein sequence as natural language input to LLMs (Wang et al., 2023b; Luo et al., 2023b; Taylor et al., 2022; Fang et al., 2024; Pei et al., 2024a). However, they focus on the learning of associations between protein sequences and their properties or functions using a small part of real-world protein sequences, which hinders the LLMs from generating reliable results for proteins at an evolutionary scale. To cope with this issue, ProtST (Xu et al., 2023) and ProteinCLAP (Liu et al., 2023) utilize pLMs pre-trained on evolutionary-

scale protein databases as protein sequences encoders and leverage contrastive learning (Radford et al., 2021b) to train on protein-text paired data, thereby incorporating functional information from text with high-quality protein representations from pLMs. Unfortunately, these methods can be solely applied to prediction and retrieval tasks about proteins whereas real-world proteins with complex and diverse properties and functions require full understanding in an open-ended generation fashion instead of these specific tasks. Therefore, Prot2Text (Abdine et al., 2024), ProteinChat (Guo et al., 2023) and ProtT3 (Liu et al., 2024b) propose the protein-to-language generation via integrating protein sequence or structure encoding from pre-trained models.

However, existing methods still show limitations in providing reliable general-purpose protein understanding for their application in scientific research. Firstly, although some studies have considered the determinant role of protein 3D structure on its properties and functions and used it as input, the proteins with directly usable 3D structural information is very rare. This leads to a situation where we need to learn the relationship between 1D sequences and functional information while relying on limited 3D information, making it challenging to provide reliable property and function predictions. Secondly, existing protein-related instruction datasets neglect 3D structures and have limited coverage of property and function types, which impedes the evaluation of model reliability and generalizability in general-purpose protein understanding tasks. Thirdly, the diversity of protein properties and functions poses a challenge for their accurate prediction. Using a single general-purpose model to predict a wide range and complex set of properties and functions is more challenging than fine-tuning multiple specialized models for different specific tasks.

To address these challenges, we propose a generalized instruction tuning framework called Structure-Enhanced Protein Instruction Tuning (SEPIT) for general-purpose protein understanding. To comprehensively evaluate the reliability and generalizability of our models, we construct the largest protein instruction dataset to date which covers the most types of protein properties and functions, based on large-scale protein knowledge bases (Bairoch & Apweiler, 1997; Varadi et al., 2022). Before instruction tuning, in order to obtain a protein sequence/structure fused encoder that supports different types of protein input (1D or 1D&3D), we specially design a structure-aware module into pLMs. Then, we warm it up through protein-text contrastive learning and structure denoising, which provides a foundation to leverage a small amount of structural information for enhancing the understanding of large-scale sequence-only proteins. After that, based on the protein sequence/structure fused encoder, we design a two-stage protein instruction tuning pipeline to enable LLMs for general-purpose protein understanding. In stage 1, we instill basic understanding of proteins into the model through protein caption instructions. In stage 2, we initialize mixture of experts through upcycling (Komatsuzaki et al., 2023), which allows the model to learn more complex and diverse functions and properties, based on the basic understanding from stage 1, without increasing additional activated parameters. In summary, our contributions include: 1) By designing a structure-aware module and integrating it into pLMs, we enable the models to handle different types of protein inputs, thereby improving the quality of embedding over the vanilla sequence-only pLMs. 2) We construct the largest and most comprehensive protein instruction dataset to date, which addresses the lack of comprehensive dataset for general-purpose protein understanding. 3) We design a two-stage protein instruction tuning pipeline that enables a single model to learn a wide range and complex set of protein properties and functions. This is achieved by leveraging Mixtures of Experts (MoEs) built upon foundational knowledge. 4) Based on the proposed SEPIT framework and protein instruction dataset, we demonstrate the feasibility of enabling LLMs with the capability of general-purpose protein understanding.

## 2 RELATED WORK

In this section, we will present the related work pertinent to our study, focusing primarily on protein language models and multimodal instruction tuning. Additionally, an introduction to related work concerning learning with 3D structural information is provided in Appendix A.

**Protein Language Models.** Employing context-aware language models (Rosenfeld, 2000), protein sequences can be likened to sentences wherein amino acids serve as the elemental words. Through pre-training on databases containing hundreds of millions of such protein sequences (*e.g.*, UniRef (Suzek et al., 2015), BFD (Steinegger et al., 2019; Steinegger & Söding, 2018)), pLMs enable effective modeling and prediction of protein structures and functions (Hu et al., 2022). In earlier works (Alley et al., 2019; Strodthoff et al., 2020; Heinzinger et al., 2019), LSTM and its variants (Hochreiter & Schmidhuber, 1997; Yu et al., 2019; Huang et al., 2015; Krause et al., 2016) were utilized to model the dependencies between residues in single protein sequences. With the rise of the Transformers

architecture (Vaswani et al., 2017), Transformers-based pLMs emerged. ESM-1b (Rives et al., 2021), leveraging the Transformers architecture along with a masking strategy for pretraining, significantly enhances the prediction accuracy for mutational effects, secondary structure, and long-range contacts. After this, ProtTrans (Elnaggar et al., 2022) released two auto-regressive models (Dai et al., 2019; Yang et al., 2019) and four auto-encoder models (Devlin et al., 2018; Lan et al., 2019; Clark et al., 2020; Raffel et al., 2020) pre-trained on protein sequence databases. Beyond merely focusing on single protein sequences, MSA-Transformer (Rao et al., 2021) integrate multiple sequence alignments (MSA) of homologous proteins, provided a solid foundation for the success of AlphaFold2 (Jumper et al., 2021). Moreover, ESM-2 (Lin et al., 2023) further scaled up pLMs, achieving protein structure prediction performance comparable to previous works without utilizing MSA information, and significantly reduced inference overhead (Lin et al., 2022). Additionally, there were other studies that attempted to incorporate additional knowledge into the pre-training of protein sequences. For instance, ProteinBERT (Brandes et al., 2022) and OntoProtein (Zhang et al., 2022) integrated gene ontology (GO) information into the representations of protein sequences, enhancing the model's understanding of protein functions. Although these pLMs can provide high-quality representations of proteins, they cannot generate natural language predictions about protein properties and functions.

**Multimodal Instruction Tuning.** With the emergence of MLLMs such as GPT4 (OpenAI et al., 2024) and Genimi (Team et al., 2023), MLLMs had become a focal point of research. Initially, works like CLIP (Radford et al., 2021b), ALBEF (Radford et al., 2021a), VLMo (Radford et al., 2021c), SimVLM (Wang et al., 2021) emphasized exploring the cross-modal alignment between vision and language. Subsequently, based on modal alignment, Flamingo (Alayrac et al., 2022) and BLIP2 (Li et al., 2023) established bridges between visual encoders and LLMs using the Perceiver Resampler and the Q-Former, respectively. Following this, PaLM-E (Driess et al., 2023) introduced "multimodal sentences" as input, injecting real-world continuous sensor data into the LLMs in the form of language tokens, thereby endowing the model with a general multi-task capability. Additionally, efforts such as InstructBLIP (Dai et al., 2024), LLaVA (Liu et al., 2024a), MiniGPT4 (Zhu et al., 2023), mPLUGOwl (Ye et al., 2023), Qwen-VL (Bai et al., 2023), CogVLM (Wang et al., 2023a) applied the crucial instruction tuning technique from LLMs to MLLMs, enhancing the MLLMs' ability to follow multimodal instructions. At the same time, they introduced innovations from the perspectives such as the construction of instruction datasets, training paradigms, and model design, which in turn refreshed the performance of MLLMs in a variety of visual-language downstream tasks (Yin et al., 2023). In this paper, we attempt to apply this paradigm to the protein domain, investigating the potential of endowing LLMs with general-purpose protein understanding capabilities.

## 3 CONSTRUCTION OF PROTEIN INSTRUCTION DATASET

To endow LLMs with general-purpose protein understanding capabilities and evaluate their reliability and generalizability, in this paper, we construct a protein instruction dataset contains open-ended generation and closed-set answer tasks. For the open-ended generation subset, we mainly constructed it based on Swiss-Prot (Bairoch & Apweiler, 1997). We include almost all protein properties and functions contained therein (Function, Similarity, Subcellular location, Induction, Molecular Function, Biological Process, Cellular Component, Developmental Stage, Short Sequence Motif, Tissue Specificity, Activity Regulation, Pathway), and used ChatGPT (OpenAI et al., 2024) to aid in designing question templates based on the structured annotations. For the closed-set answer subset, we constructed it mainly based on the RCSB PDB (RCSB, 2024). We follow the data organized by previous researchers (Guo et al., 2023) and select parts of their proposed Q&A samples that are highly related to protein properties and functions, filtering out other samples related to metadata (*e.g.* discovery time and discovery methods). We have also sampled some examples related to Enzyme Commission (EC) and Gene Ontology (GO) predictions (Gligorijević et al., 2021) for inclusion in the closed-set answer subset. More detailed information about the dataset, including statistical information and examples, are shown in Appendix B.

Compared to previously proposed protein-text related datasets (Xu et al., 2023; Fang et al., 2024; Wang et al., 2023b; Guo et al., 2023), our advantages are as follows: First, our dataset contains the most comprehensive set of instructions, covering almost all critical protein properties and function types found in databases (Bairoch & Apweiler, 1997). Second, our dataset includes the largest volume of instructions, comprising over 10 million instructions (with an additional 5 million supplementary instructions from TrEMBL). Third, our dataset incorporates structural information, offering the

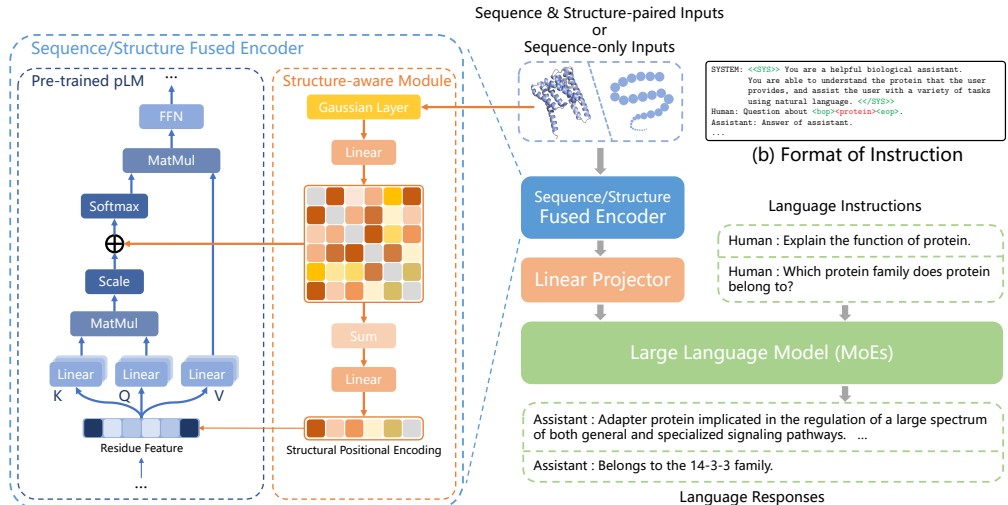

Figure 1: (a) The model architecture of the SEPIT framework includes sequence/structure fused protein encoder, linear projector, and LLMs with MoEs modules, (b) example of instruction format.

potential to enhance prediction reliability by leveraging this structural data. In summary, our dataset has the potential to support future research in the field of general-purpose protein understanding.

## 4 STRUCTURE-ENHANCED PROTEIN INSTRUCTION TUNING

In this section, we provide a detailed introduction to our proposed SEPIT framework from two perspectives: model architecture and training pipeline.

### 4.1 MODEL ARCHITECTURE

In this subsection, we introduce three main components of the SEPIT framework: 1) a sequence/structure fused protein encoder, 2) a linear projector, and 3) a large language model with mixture of experts modules. The illustration of the model architecture is depicted in Figure 1.

**Sequence/Structure Fused Protein Encoder.** Considering the abundance of sequence-only data and the relatively small amount of sequence-structure paired data (whether experimentally-determined structures or computed structures) within the protein domain (Varadi et al., 2022; Steinegger et al., 2019; Steinegger & Söding, 2018; Suzek et al., 2015), we propose a Sequence/Structure Fused Protein Encoder that is capable of accommodating inputs in either form. Additionally, through leveraging the limited sequence-structure paired data, we aim to enhance the model's performance when dealing with sequence-only inputs.

For sequence-only data, numerous pLMs (Rives et al., 2021; Lin et al., 2023; Brandes et al., 2022; Elnaggar et al., 2022; Rao et al., 2021) have already been pre-trained on it. To enable them to perceive structural information, we have designed structure-aware modules for pLMs. Mainstream pLMs, such as ESM (Rives et al., 2021; Lin et al., 2023), are encoder-only architectures, consisting of multiple Transformer encoder layers, which primarily comprise self-attention modules (Vaswani et al., 2017) and feed-forward network (FFN). Here, our main focus is on the self-attention modules (for simplicity, we discuss the scenario with single-head and assume that the dimensions of the query, key, and value are all equal to the hidden size $d$). Let $\boldsymbol{X}^{(l)} = [\boldsymbol{x}_1^{(l)}, \boldsymbol{x}_2^{(l)}, \cdots, \boldsymbol{x}_N^{(l)}]^\top$ denote the input to self-attention module in $l$-th layer, where $\boldsymbol{x}_i^{(l)} \in \mathbb{R}^d$ is the $d$-dimension representation of the $i$-th residue out of the $N$ residues in the protein. The self-attention module then works as follows:

$$\boldsymbol{A}^{(l)} = \frac{\boldsymbol{X}^{(l)}\boldsymbol{W}_Q^{(l)}(\boldsymbol{X}^{(l)}\boldsymbol{W}_K^{(l)})^\top}{\sqrt{d}}, \tag{1}$$

$$\text{Attn}(\boldsymbol{X}^{(l)}) = \text{softmax}(\boldsymbol{A}^{(l)})\boldsymbol{X}^{(l)}\boldsymbol{W}_V^{(l)}, \tag{2}$$

where $\boldsymbol{W}_Q^{(l)} \in \mathbb{R}^{d\times d}, \boldsymbol{W}_K^{(l)} \in \mathbb{R}^{d\times d}, \boldsymbol{W}_V^{(l)} \in \mathbb{R}^{d\times d}, \boldsymbol{A}^{(l)}$ is the attention matrix, $\boldsymbol{A}_{i,j}^{(l)}$ denotes the similarity between residue $i$ and $j$. Inspired by previous work on geometric Transformers (Zhou et al., 2023; Luo et al., 2023a), our structure-aware module takes the 3D coordinates

$\boldsymbol{C} = [\boldsymbol{c}_1, \boldsymbol{c}_2, \cdots, \boldsymbol{c}_N]^\top$, $\boldsymbol{c} \in \mathbb{R}^3$ of residues (alpha carbon atoms) as input and outputs the 3D feature matrix $\boldsymbol{\Delta} \in \mathbb{R}^{N \times N}$, representing the pairwise spatial relationships of residues in 3D space:

$$\boldsymbol{\Delta} = \phi\left(\boldsymbol{\psi}_{(i,j)} \boldsymbol{W}_a\right) \boldsymbol{W}_b, \qquad (3)$$

where $\boldsymbol{W}_a \in \mathbb{R}^{K \times K}$, $\boldsymbol{W}_b \in \mathbb{R}^{K \times 1}$ are linear transformation and $\boldsymbol{\psi}_{(i,j)} = [\psi^1_{(i,j)}, \cdots, \psi^K_{(i,j)}]^\top$ is the Euclidean distance for each twosome of residues undergoes a transformation via the Gaussian Basis Kernel function (Scholkopf et al., 1997):

$$\psi^k_{(i,j)} = -\frac{1}{\sqrt{2\pi}|\sigma^k|} \exp\left(-\frac{1}{2}\left(\frac{\|\boldsymbol{c}_i - \boldsymbol{c}_j\| - \mu^k}{|\sigma^k|}\right)^2\right), \ k = 1, ..., K, \qquad (4)$$

where the learnable parameters $\mu^k$ and $\sigma^k$ correspond to the center and scaling coefficient of the $k$-th Gaussian Basis Kernel. These relationships are incorporated into the attention matrix as bias:

$$\hat{\boldsymbol{A}}^{(l)} = \boldsymbol{A}^{(l)} + \boldsymbol{\Delta}, \qquad (5)$$

and added as structure positional encoding to the embedding input of pLMs:

$$\hat{\boldsymbol{X}}^{(0)} = \boldsymbol{X}^{(0)} + \omega\left(\sum_{j \in [n]} \boldsymbol{\psi}_{(i,j)}\right) \boldsymbol{W}_c, \qquad (6)$$

where $\omega$ signifies the coefficient that regulates the magnitude of the structure positional encoding and $\boldsymbol{W}_c \in \mathbb{R}^{K \times d}$ is learnable linear transformation. It is noteworthy that when the input to the sequence/structure fused protein encoder consists solely of the protein sequence, lacking structural information, the structure-aware module will be automatically disabled. This allows it to adapt to different types of protein inputs.

**Linear Projector.** To bridge proteins with natural language, a module is required to link the protein encoder and the LLMs decoder. Prior work in the MLLMs field has contributed outstanding methods such as Q-former (Li et al., 2023), linear projector (Liu et al., 2024a), and merging tokens before the linear projector (Zhu et al., 2023). Considering the vast differences between proteins and visual images - that is, the former requires the retention of more information of all residues (as any change in the amino acid sequence can lead to significant structural differences, resulting in profoundly different properties and functions), whereas the latter possesses some degree of information redundancy - we opt for a simple linear projector to reduce information loss.

**Large Language Model with Mixture of Experts Module.** Due to the understanding of proteins being a complex multi-task problem, the various properties and functions of proteins can exhibit significant changes with subtle variations in the amino acid sequence. As the hub of "understanding" within the entire framework, the capabilities of LLMs are crucial. Previous scaling laws (Kaplan et al., 2020) have suggested that larger parameter sizes can endow LLMs with stronger capabilities; however, the additional computation costs brought about by increased activated parameters are intolerable for us. Therefore, we seek to leverage mixture-of-experts (MoEs) to achieve higher parameter sizes without increasing the number of activated parameters, thereby enhancing the model's capacity and generalization ability. In our framework, the MoEs module replaces the FFN module in each Transformer decoder layer. The MoEs module works as follows (Lepikhin et al., 2020; Jacobs et al., 1991; Zoph et al., 2022; Lin et al., 2024):

$$y = \sum_{i=1}^{n} \mathrm{G}(\boldsymbol{x})_i \cdot \mathrm{E}_i(\boldsymbol{x}), \qquad (7)$$

Here, $\boldsymbol{x}$ is assumed to be the input to the original FFN layer, and E represents the $n$ experts in the MoEs, each of which has the exact same structure as the original FFN layer. G represents the gating network, $\mathrm{G}(\boldsymbol{x})_i$ denotes the gate weight for the $i$-th expert, and $\mathrm{E}_i(\boldsymbol{x})$ is the output of the $i$-th expert. For the gating network, we employed the commonly used linear TopK gate:

$$\mathrm{G}(\boldsymbol{x}) := \mathrm{Softmax}\left(\mathrm{TopK}\left(\boldsymbol{x} \cdot \boldsymbol{W}_{\mathrm{G}}\right)\right), \qquad (8)$$

and we imposed auxiliary loss to ensure the token balance among the experts (Zoph et al., 2022; Lepikhin et al., 2020):

$$\mathcal{L}_{\mathrm{aux}} = n \cdot \sum_{i=1}^{n} f_i \cdot P_i, \qquad (9)$$

where $f_i$ is the proportion of tokens processed by expert $i$ and $p_i$ represents the proportion of gating weight allocated to an expert.

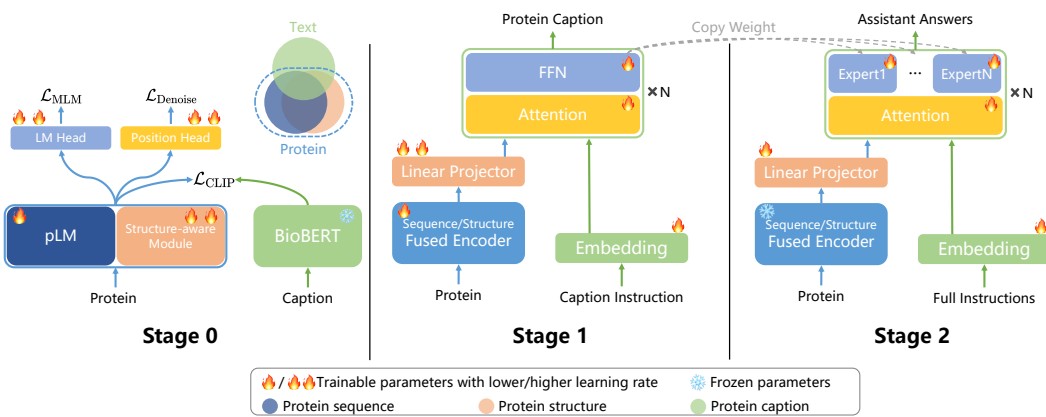

Figure 2: The three-stage training pipeline of SEPIT with a warm-up stage (Stage 0) for protein encoder, and a two-stage instruction tuning (Stage 1 & Stage 2).

## 4.2 TRAINING PIPELINE

In this subsection, we will discuss the details of the training pipeline for SEPIT framework, based on the model architecture presented before. As shown in Figure 2, the whole pipeline includes three stages: in Stage 0, we warm up our proposed sequence/structure fused protein encoder based on pre-trained pLM primarily through protein-text contrastive learning and structure denoising. In Stage 1, we pre-train on the protein captioning task to further align protein representations with natural language, while concurrently infusing foundational protein knowledge into LLMs. In Stage 2, we initiate the MoEs modules using the sparse upcycling (Komatsuzaki et al., 2023) from the FFNs of the LLMs trained in Stage 1 and perform instruction tuning on our proposed protein instruction dataset.

**Stage 0: Warming Up the Sequence/Structure Fused Protein Encoder.** In this stage, our primary goal is to warm up our protein encoder. Although the pLM is already pre-trained, the structure-aware module we plug in is randomly initialized. To address this, we leverage two main pre-training paradigms. Firstly, to enable the structure-aware module to better perceive structural information, we follow the common practice in molecular self-supervised learning (Godwin et al., 2022; Zaidi et al., 2023; Luo et al., 2023a; Zhou et al., 2023), which involves structure denoising tasks. For the input 3D coordinates of protein residues $C = [c_1, c_2, \cdots, c_N]^\top$, we apply noise to obtain nosied coordinates $\tilde{C} = [\tilde{c}_1, \tilde{c}_2, \cdots, \tilde{c}_N]^\top$, where $\tilde{c}_i = c_i + \alpha\delta_i$, $\delta_i \sim \mathcal{N}(0, I)$, $\alpha$ is a scaler used to control the magnitude of noise. Then we predict the applied noise based on them (equivalent to predicting the original 3D coordinates). The formula for denoise loss is as follows:

$$\mathcal{L}_{\text{Denoise}} = \frac{1}{3n} \sum_{i=1}^{n} \sum_{j=1}^{3} \left(\delta_i^j - \hat{\delta}_i^j\right)^2, \tag{10}$$

where $\hat{\delta}_i^j$ is output by an additional SE(3) equivariant attention layer (Luo et al., 2023a; Shi et al., 2023) (position head), which takes the last hidden state of the protein encoder $X^{(L+1)}$ and $\Delta$ in Equation 5 as input:

$$\hat{\delta}_i^j = \left(\sum_{i=1}^{n} \Delta_{ik} D_{ik}^j X_k^{(L+1)} W_m\right) W_n, \quad D_{ik} = \frac{c_i - c_k}{\|c_i - c_k\|}. \tag{11}$$

Secondly, we utilize protein-text contrastive learning to further promote the fusion of protein sequence and structure information under text supervision, while concurrently aligning protein representations with their textual descriptions. Formally, given a batch of paired proteins and protein captions $\{(\mathcal{P}_i, \mathcal{T}_i)\}_{i=1}^{B}$, the CLIP loss can be expressed as:

$$\mathcal{L}_{\text{CLIP}} = -\frac{1}{2B} \sum_{i=1}^{B} \left(\log \frac{\exp(p_i \cdot t_i/\tau)}{\sum_{j=1}^{B} \exp(p_i \cdot t_j/\tau)} + \log \frac{\exp(p_i \cdot t_i/\tau)}{\sum_{j=1}^{B} \exp(p_j \cdot t_i/\tau)}\right), \tag{12}$$

where $p_i, t_i$ are the representations of $P_i$ and $T_i$ output by the protein encoder and text encoder (BERT), respectively. Additionally, we also maintain the Masked Language Model (MLM) training objective related to the protein sequence (consistent with ESM2 (Lin et al., 2023)) as a regularization term to prevent catastrophic forgetting in the pLM. Overall, the loss for Stage 0 is as follows:

$$\mathcal{L}_{\text{Stage 0}} = \mathcal{L}_{\text{Denoise}} + \mathcal{L}_{\text{CLIP}} + \mathcal{L}_{\text{MLM}}. \tag{13}$$

Under the influence of these training objectives, the mutual information between protein sequence and protein structure as well as protein and text is increased.

**Stage 1: Pre-training on Protein Captions.** In this stage, our fundamental objective is to further align proteins with their natural language descriptions through the paradigm of conditional generation (Liu et al., 2024a; Ouyang et al., 2022), utilizing protein caption instructions. To ensure consistency in the model's handling of different forms of protein inputs, we randomly input protein data, both those with only sequences and those paired with structures, into our protein encoder at probabilities of $15\%$ and $85\%$, respectively. The output protein representation sequences are mapped to the textual space through linear projector, in conjunction with protein caption instructions to guide LLMs in producing straightforward descriptions of proteins, such as function, family, subcellular localization, and overall descriptions. Formally, consider a protein-text pair $(\mathcal{P}, \mathcal{T})$ similar to that in Stage 0, given the output sequence of the protein encoder $\boldsymbol{S_p}$, and instructions $\boldsymbol{S}_{\text{instruct}}$, the objective of Stage 1 is as follows:

$$\mathcal{L}_{\text{Stage 1}} = p(\boldsymbol{S_t}|f(\mathbf{S_P}), \boldsymbol{S}_{\text{instruct}}) = \prod_{i=1}^{L} p_{\boldsymbol{\theta}}(s_i|\boldsymbol{S_p}, \boldsymbol{S}_{\text{instruct}, <i}, \boldsymbol{S_{t}, <i}), \quad (14)$$

where $f(\cdot)$ denotes the linear projector, $\theta$ represents all trainable parameters, $\boldsymbol{S}_{\text{instruct}, <i}$ and $\boldsymbol{S_{t}, <i}$ respectively signify the instruction and answer tokens preceding the current prediction token $s_i$, and $L$ denotes the total sequence length accepted by LLMs.

**Stage 2: Upcycling and Instruction Tuning.** In this stage, our main aim is to upcycle the model obtained from Stage 1 by replacing each FFN module within the LLMs with MoE module, where each expert is initialized by an FFN. In the case of top$-1$ activation, this approach offers a larger model parameter count under the same activation parameter volume. Meanwhile, the basic understanding of proteins acquired in Stage 1 lays the groundwork for more complex and multifaceted learning in this stage. Based on this, we utilize a diverse set of protein instructions for instruction tuning, aiming to endow SEPIT with general-purpose protein understanding capabilities. Similar to Stage 1, the loss function for Stage 2 can be formally represented as:

$$\mathcal{L}_{\text{Stage 2}} = \mathcal{L}_{\text{Stage 1}} + \beta \mathcal{L}_{\text{aux}}, \quad (15)$$

where $\mathcal{L}_{\text{aux}}$ is an auxiliary loss used for constraining the token balance among experts, as mentioned in Equation 9, with $\beta$ being used to control its relative magnitude.

## 5 EXPERIMENTS

In this section, we will first introduce the experimental setting in this paper. Subsequently, we will comprehensively demonstrate the effectiveness of SEPIT and its various designs through performance comparisons and ablation studies. Finally, we will delve deeper into the characteristics of SEPIT through case studies.

### 5.1 EXPERIMENTAL SETTING

First, we provide a brief overview of the main settings in experiments. More detailed information such as implementation details and dataset construction details can be found in Appendix C.1 and B.

**Use of Pre-training Data.** In each stage of SEPIT, we utilize our proposed protein instruction dataset, employing different subsets at various stages. At Stage 0, our focus is primarily on basic protein descriptions derived from Swiss-Prot and the RCSB PDB. For Swiss-Prot, akin to previous work (Xu et al., 2023), we formulate the protein captions using functions, subcellular locations, and similarities. Regarding the RCSB PDB, we directly utilize abstracts from related PubMed papers collected by (Guo et al., 2023) as captions. In Stage 1, we employee the same data as in Stage 0, but the output format is altered to the style of caption instructions. During Stage 2, we utilize the complete protein instruction dataset we proposed, which includes open-ended generation and closed-set answers tasks.

**Baselines and Evaluation Metrics.** We evaluate the capability of SEPIT for general-purpose protein understanding on the test set of the protein instruction dataset we proposed with the state-of-the-art models. There are four main categories of methods. Among the Zero-Shot methods, we include current mainstream LLMs providing API services (e.g., Claude-3-haiku (Anthropic, 2024), GPT-3.5-turbo (OpenAI, 2024a), and GPT-4-turbo (OpenAI, 2024b)), open-source LLMs fine-tuned on biomedical corpus (e.g., Galactica (Taylor et al., 2022), BioMedGPT (Luo et al., 2023b)) and open-source LLMs fine-tuned specifically on molecular or protein knowledge(e.g., Mol-Instructions (Fang

Table 1: Performance comparisons on open-ended generation and closed-set answer tasks.

| Model | Activated Parameters | Open-ended | | | | | | | | | Closed-set |
|---|---|---|---|---|---|---|---|---|---|---|---|
| | | BLEU-2 | BLEU-4 | ROUGE-1 | ROUGE-2 | ROUGE-L | METEOR | BERT-P | BERT-R | BERT-F1 | Accuracy |
| **Zero-Shot** | | | | | | | | | | | |
| GPT-3.5-turbo | N/A | 3.26 | 0.02 | 12.41 | 3.14 | 11.06 | 10.44 | 85.18 | 85.40 | 85.24 | 56.56% |
| Claude-3-haiku | N/A | 3.00 | 0.07 | 12.10 | 2.65 | 10.62 | 9.28 | 86.04 | 85.47 | 85.70 | 59.14% |
| GPT-4-turbo | N/A | 4.21 | 0.08 | 12.78 | 2.93 | 11.57 | 10.41 | 86.91 | 85.56 | 85.71 | 58.58% |
| Galactica | 1.3B | 0.43 | 0.01 | 3.49 | 0.41 | 2.67 | 2.44 | 85.79 | 82.61 | 84.08 | 39.15% |
| BioMedGPT | 7B | 0.83 | 0.01 | 4.90 | 0.49 | 3.26 | 4.59 | 85.51 | 84.95 | 85.14 | 38.61% |
| Mol-Instructions | 7B | 0.53 | 0.01 | 5.96 | 0.39 | 4.64 | 5.51 | 83.81 | 84.41 | 84.06 | — |
| BioT5+ | 252M | 3.88 | 1.92 | 12.12 | 4.88 | 10.37 | 14.26 | 85.14 | 85.93 | 85.48 | — |
| InstructProtein | 1.3B | 5.50 | 2.97 | 14.80 | 5.68 | 13.76 | 13.17 | 85.34 | 85.92 | 85.57 | 48.37% |
| **Instruction Tuning** | | | | | | | | | | | |
| TinyLlama | 1.1B | 51.16 | 43.44 | 65.41 | 51.26 | 62.31 | 60.80 | 93.97 | 94.37 | 94.16 | 74.09% |
| OpenLlama-v2 | 3B | 36.19 | 30.65 | 48.33 | 36.52 | 45.53 | 49.01 | 92.92 | 91.87 | 92.35 | 71.77% |
| Llama2 | 7B | 57.02 | 49.47 | 70.80 | 57.24 | 67.78 | 65.96 | 94.95 | 95.17 | 95.05 | 71.68% |
| **Sequence-Only Protein Instruction Tuning** | | | | | | | | | | | |
| PIT-TinyLlama | 1.8B | 57.82 | 50.02 | 71.34 | 58.16 | 68.35 | 66.19 | 95.18 | 95.28 | 95.26 | 76.02% |
| PIT-TinyLlama-MoEs | 1.8B | 57.92 | 50.01 | 72.13 | 58.21 | 69.19 | 66.29 | 95.31 | 95.30 | 95.29 | 78.56% |
| **Structure-Enhanced Protein Instruction Tuning** | | | | | | | | | | | |
| SEPIT-TinyLlama | 1.8B | 58.43 | 51.04 | 72.34 | 58.77 | 69.13 | 67.91 | 95.32 | 95.59 | 95.44 | 79.05% |
| **SEPIT-Llama2** | 8B | **60.81** | **52.37** | **74.80** | **60.84** | **71.62** | **68.43** | 95.81 | **95.73** | **95.76** | **79.97%** |
| **SEPIT-TinyLlama-MoEs** | 1.8B | 60.28 | 52.16 | 74.22 | 60.29 | 71.13 | 68.27 | 95.62 | 95.69 | 95.64 | 79.73% |

et al., 2024), BioT5+ (Pei et al., 2024a) and InstructProtein (Wang et al., 2023b)). In the category of instruction tuning methods, we evaluate mainstream open-source LLMs (e.g., TinyLlama-Chat (Zhang et al., 2024), OpenLlama-v2 (OpenLLMAI, 2023), Llama2-Chat (Touvron et al., 2023)), where the protein sequences are input in natural language form. For sequence-only protein instruction tuning methods (PITs), ESM2-650M (Lin et al., 2023) was utilized as the protein encoder, with only protein sequences input. In addition to our proposed SEPIT framework, SEPIT-TinyLlama-MoEs, we have also designed two variants that differ in the LLMs' architecture. For evaluation metrics, we employ BLEU score (Sutskever et al., 2014), ROUGE score (Lin, 2004), METEOR score (Banerjee & Lavie, 2005), BERT score (Zhang et al., 2019) calculated by PubMedBERT (Gu et al., 2021), and Accuracy to assess performance across two types of tasks: open-ended generation and closed-set answer, respectively. It is worth noting that related works (Lv et al., 2024; Guo et al., 2023; Liu et al., 2024b) exist that can caption protein sequences but lack instruction-following abilities. Direct comparison with these methods is not possible here. However, we have compared representative methods of them using an alternative approach detailed in the Appendix C.3.

## 5.2 PERFORMANCE COMPARISONS

The results of performance comparisons are shown in Table 1. We can observe that: 1) Our proposed SEPIT consistently outperforms the baseline models by a significant margin. Specifically, SEPIT-Llama achieves the highest performance across all metrics in both types of tasks. In comparison, SEPIT-TinyLlama-MoEs demonstrates significantly higher parameter efficiency, achieving almost identical results to SEPIT-Llama with just 1/6 of the LLMs' activated parameters. 2) Zero-Shot methods generally perform poorly, with neither powerful general models like GPT and Claude nor open-source models fine-tuned on biomedical corpus or protein knowledge able to accomplish protein understanding tasks well. Notably, Mol-Instructions and BioT5+ are trained on protein-related instructions. However, limited data diversity causes catastrophic forgetting to instruction following, hindering their ability to provide closed-set answers or accurate open-ended responses. 3) Instruction tuning on pure LLMs endows LLMs with decent protein understanding capabilities and demonstrates certain scaling laws (Kaplan et al., 2020) (OpenLlama-v2 demonstrates suboptimal results as it has not been specifically fine-tuned for chat assistant purpose.) However, overall, due to the lack of prior knowledge learned from evolutionary-scale protein databases, they can only achieve limited performance. 4) While utilizing prior knowledge from pLMs can significantly enhance the performance of LLMs of the same scale, the lack of structural awareness in PIT only results in suboptimal outcomes compared to our proposed SEPIT.

## 5.3 ABLATION STUDY

In this section, we will explore the impact of various designs of SEPIT on its performance from two aspects: the model and the data.

**Model Ablation.** For SEPIT's model architecture, we propose the following variants: without the structure-aware module (w/o Structure), without the mixture of experts module (w/o MoEs), without both the aforementioned modules (w/o Structure & MoEs), without Stage 0 pre-training (w/o Stage 0),

and completely excluding the SEPIT framework (w/o SEPIT). Table 2 shows the results of the ablation study, proving the significant effectiveness of each component within SEPIT's model architecture. The absence of either the structure-aware module or the MoEs Module leads to performance degradation in both open-ended generation tasks and closed-set answer tasks, with further deterioration when both are excluded. Meanwhile, the performance under w/o SEPIT intuitively demonstrates the overall effectiveness of the framework. It is noteworthy that the results for w/o Stage 0 are not available, as under the implementation using automatic mixed precision (AMP) based on FP16, the randomly initialized structure-aware module would bring excessively large gradients causing overflow, even though we employed a warm-up learning rate scheduler. Due to device restrictions, we were unable to use BF16 type; however, this issue was resolved as Stage 0 progressed. In order to supplement the analysis of the effectiveness of Stage 0, we validate the protein encoder pre-trained by Stage 0 on commonly used EC, GO annotation tasks. (Gligorijević et al., 2021). The results, as shown in Table 3, demonstrates its superior performance compared to state-of-the-art methods on $F_{max}$.

**Data Ablation.** Regarding the data, apart from using Swiss-Prot and RCSB PDB for constructing the protein instruction dataset, there exists a substantial amount of protein-text paired data in TrEMBL (Bairoch & Apweiler, 1997). Considering that the TrEMBL data is annotated by automated methods and has not been manually screened, we select proteins with more comprehensive descriptions (annotation score $\geq 4$) to construct a supplementary dataset using the same method, with a sample size (5.25M) close to the entire protein instruction dataset (5.47M). Disappointingly, as shown in Table 2 (w/ TrEMBL), even after doubling the GPUs cost, what we obtain is a decrease in performance. Our analysis suggests that directly mixing low-quality data into high-quality data introduces noise, and protein understanding tasks require higher quality over quantity for data. More results can be found in Appendix C.4.

Table 2: Ablation study on SEPIT's architecture.

| Model | Open-ended | | | Closed-set |
|---|---|---|---|---|
| | BLEU-2 | ROUGE-L | METEOR | Accuracy |
| SEPIT-TinyLlama-MoEs | **60.28** | **71.13** | **68.27** | **79.73%** |
| w/o Structure | ↓ 4.08% | ↓ 2.81% | ↓ 2.98% | ↓ 1.48% |
| w/o MoEs | ↓ 3.17% | ↓ 2.90% | ↓ 0.52% | ↓ 0.86% |
| w/o Structure & MoEs | ↓ 4.26% | ↓ 4.07% | ↓ 3.13% | ↓ 4.88% |
| w/o Stage 0 | — | | | |
| w/o SEPIT | ↓ 17.83% | ↓ 14.17% | ↓ 12.28% | ↓ 7.61% |
| w/ TrEMBL | ↓ 2.69% | ↓ 2.13% | ↓ 2.00% | ↓ 0.26% |

Table 3: Performance of SEPIT's encoder.

| Model | EC | GO | | |
|---|---|---|---|---|
| | | BP | MF | CC |
| ProtBert (Brandes et al., 2022) | 0.838 | 0.279 | 0.456 | 0.408 |
| OntoProtein (Zhang et al., 2022) | 0.841 | 0.436 | 0.631 | 0.441 |
| ESM1b (Rives et al., 2021) | 0.869 | 0.452 | 0.659 | 0.477 |
| ESM2 (Lin et al., 2023) | 0.874 | 0.472 | 0.662 | 0.472 |
| CDConv (Fan et al., 2023) | 0.820 | 0.453 | 0.654 | 0.479 |
| GearNet (Zhang et al., 2023b) | 0.810 | 0.400 | 0.581 | 0.430 |
| ProtST-ESM2 (Xu et al., 2023) | 0.878 | **0.482** | 0.668 | 0.487 |
| **SEPIT's Encoder** | **0.893** | 0.476 | **0.674** | **0.497** |

## 5.4 CASE STUDY

**Consistency of SEPIT with Different Protein Input Formats.** Towards a general-purpose protein understanding capability, SEPIT supports both sequence-only and sequence-structure paired protein inputs, achieving consistent results as shown in Table 4. The three SEPIT variants all yields very similar effects on both types of protein inputs. Moreover, compared to the corresponding scale PIT model, SEPIT demonstrates a stronger understanding of sequence-only inputs. This implies that through SEPIT, we can utilize a small amount of sequence-structure paired data to enhance the understanding of a large volume of sequence-only protein inputs.

Table 4: Performance of SEPIT with different protein input formats.

| Model | Train w/ Struct. | Infer w/ Struct. | Open-ended Generation | | | | Closed-set Answer |
|---|---|---|---|---|---|---|---|
| | | | BLEU-2 | ROUGE-L | METEOR | BERT-F1 | Accuracy |
| PIT-TinyLlama | ✗ | ✗ | 57.82 | 68.35 | 66.19 | 95.26 | 76.02% |
| PIT-TinyLlama-MoEs | ✗ | ✗ | 57.92 | 69.19 | 66.29 | 95.29 | 78.56% |
| SEPIT-TinyLlama | ✓ | ✓ | 58.43 | 69.13 | 67.91 | 95.44 | 79.05% |
| | | ✗ | 57.95 | 68.75 | 67.54 | 95.38 | 77.80% |
| SEPIT-Llama | ✓ | ✓ | 60.81 | 71.62 | 68.43 | 95.76 | 79.97% |
| | | ✗ | 60.64 | 71.48 | 68.34 | 95.74 | 79.91% |
| SEPIT-TinyLlama-MoEs | ✓ | ✓ | 60.28 | 71.13 | 68.27 | 95.64 | 79.73% |
| | | ✗ | 59.98 | 70.87 | 68.00 | 95.59 | 79.53% |

Table 5: Case studies on general-purpose protein understanding ability of SEPIT.

| | |
|---|---|
| P9WF36 (Swiss-Prot) | **Question:** Explain the function of <protein>.
**Ground Truth:** Acts as a transcriptional regulator. Probably redox-responsive. The apo- but not holo-form probably binds DNA (By similarity).
**SEPIT-TinyLlama-MoEs:** Acts as a transcriptional regulator. Probably redox-responsive. The apo- but not holo-form probably binds DNA.
**PIT-TinyLlama-MoEs:** Acts as a transcriptional regulator, probably involved in antibiotic resistance gene expression. Binds DNA (By similarity).
**Llama-Chat:** Involved in resistance toward heavy metals (PubMed:12839546). Binds cadmium or lead ions but not zinc ions (By similarity). May be involved in intracellular transport of Cd(+) or Pb(+), as well as their detoxification by sequestration into vacuoles (Probable) (PubMed:17081065).
**GPT-4-turbo:** The string you've provided appears to be an amino acid sequence of a protein. However, without additional context or data, I cannot provide a function caption for this specific sequence directly. |
| A0KR49 (Swiss-Prot) | **Question:** Which GO terms outline the biological processes of <protein>?
**Ground Truth:** fatty acid beta-oxidation
**SEPIT-TinyLlama-MoEs:** fatty acid beta-oxidation
**PIT-TinyLlama-MoEs:** fatty acid beta-oxidation; phenylacetate catabolic process
**Llama-Chat:** fatty acid beta-oxidation; phenylacetate catabolic process
**GPT-4-turbo:** To understand the biological processes of the provided protein sequence, one must first consider its origin, functions, and structure. However, without direct access to databases or running bioinformatic tools right now... |

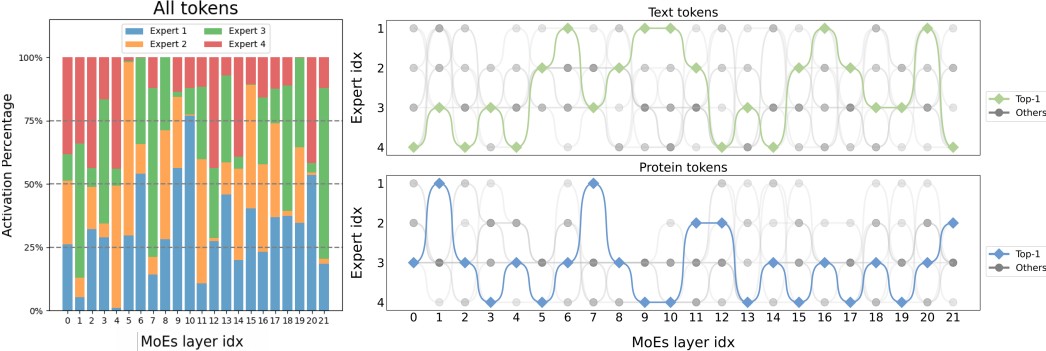

Figure 3: The workload of experts in SEPIT (left) and tokens' pathways among experts (right).

**Workload of Experts in SEPIT.** In SEPIT, we utilize mixture of experts, and in Figure 3, we present the workload distribution of different experts during inference on test set of the open-ended generation task. In the left graph, we can observe that the experts are evenly activated, indicating that the auxiliary loss has played its expected role, which lays the foundation for efficient parallel inference of experts. In the right graph, we visualize the pathways distribution of text and protein tokens across experts in different layers, and we have observed an intriguing phenomenon. Unlike the findings in previous vision-language multimodal research (Lin et al., 2024), in SEPIT, text and protein tokens are processed by different experts instead of following almost identical pathways as do text and image tokens. We believe that this stems from the fundamental difference between proteins and images. That is, protein tokens, which represent amino acids, cannot reflect the properties of the entire protein, while image tokens represent specific regions of an image containing independent information that can correspond to a part of the image's caption. This validates our choice to use complete protein representation sequences as inputs for LLMs, rather than compressing tokens as is often done in vision-language tasks. More visualization is shown in Appendix C.5.

**General-Purpose Protein Understanding Ability of SEPIT.** At last, we would like to showcase the general-purpose protein understanding capabilities of SEPIT. As shown in Table 5, we present two cases from the protein instruction dataset's test. For case 1, SEPIT accurately responds regarding the protein's function, whereas PIT incorporates incorrect details, and both Llama-Chat and GPT-4 offer entirely inaccurate responses. For case 2, SEPIT also gives the correct response, while the answers from PIT and Llama-Chat, although covering the correct answer, come with additional incorrect information, likely due to hallucinations caused by the lack of structural information. Due to space limitations, more cases are included in the Appendix C.5.

## 6 CONCLUSION

In this work, we introduce SEPIT, a novel approach for general-purpose protein understanding. SEPIT aims to enable LLMs to interpret both the sequence and structural information of proteins, thus following instructions to generate specific understanding on protein properties and functions. To achieve this, we integrate structure-aware enhancements into pre-trained pLM and connect them to LLMs via a linear projector. The models are then trained in a two-stage instruction tuning pipeline on protein instruction dataset we constructed which is the largest protein instruction dataset to date. Experimental results show that SEPIT significantly outperforms the state-of-the-art models.

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

APPENDIX

# A  SUPPLEMENT TO THE RELATED WORK

**Learning with 3D Structural Information.**    Although pLMs pre-trained on protein sequences have been proven to be effective in numerous tasks, the protein structure is inherently a determinant of protein function (Anfinsen & Haber, 1961; Hartley, 1951). More effectively utilizing 3D structural information can help to understand proteins more comprehensively.  To capture the impact of geometric positioning and interaction relationships among residues on protein's function, a class of methods encoded 3D geometric information into rotation-invariant scalars, which was then processed through graph neural networks (GNNs) for message passing (Wu et al., 2022; Zhang et al., 2023a). For example, IEConv(Hermosilla et al., 2021) utilized a multi-graph to depict primary and secondary structures through covalent and hydrogen bonds and represented the tertiary structure with the spatial 3D coordinates of atoms. By blending intrinsic and extrinsic node distances and employing hierarchical pooling, it effectively perceived all three structural levels of proteins. GearNet went further by incorporating three types of directed edges (sequential edges, radius edges, and k-NN edges) into the graph, capturing information at various structural levels. On this basis, CDConv (Fan et al., 2023) innovatively utilized MLP to parameterize the kernel matrices, as opposed to employing distinct kernel matrices for varying edge types, which enabled a more flexible and efficient modeling of complex interactions between residues. Additionally, another class of methods sought to prevent the loss of 3D structural information by incorporating 3D rigid transformations into the network operations. This led to the development of geometric GNNs/Transformers characterized by SE(3) invariance and equivariance (Liao & Smidt, 2022; Liao et al., 2024; Fuchs et al., 2020; Zhou et al., 2023; Luo et al., 2023a; Wang et al., 2024).  Representative examples of this approach, such as GVP (Jing et al., 2021) and EvoFormer (Jumper et al., 2021), were utilized by ESM-IF (Hsu et al., 2022), AlphaFold2 (Jumper et al., 2021), respectively. Furthermore, to leverage information from both evolutionary-scale protein sequences and the relatively limited protein structures, some other methods (Wang et al., 2022; Zhang et al., 2023a) attempted to build a bridge between these two domains.

## B CONSTRUCTION OF PROTEIN INSTRUCTION DATASET

Currently, the widely recognized protein-text paired databases in the protein domain mainly include Swiss-Prot, TrEMBL (Bairoch & Apweiler, 1997), and RCSB PDB (RCSB, 2024), with their specific contents detailed in Table 6. Since most of the text content in TrEMBL comes from automatic annotation methods, to eliminate the impact of its unreliability on the main experiments, unless otherwise specified, this paper only uses protein-text information from Swiss-Prot and RCSB PDB databases. The corresponding structures are from AlphafoldDB (Varadi et al., 2022) and experimentally determined structures in RCSB PDB. The statistical information about the complete protein instruction dataset is shown in Table ,7 and Figure 4.

Table 6: Protein-text paired database.

| Database | Content of Related Text | # Protein | Structure |
|---|---|---|---|
| Swiss-Prot | Manually calibrated structured annotations | 571,282 | AlphafoldDB |
| TrEMBL | Automatic structured annotation | 248,234,451 | N/A |
| RCSB PDB | Publication about protein / Meta data of protein | 204,826 | Experimentally-determined |

Table 7: Statistical information about the protein instruction dataset.

| Response Type | Data Source | Question Type | # Train Instructions | # Test Instructions |
|---|---|---|---|---|
| Open-ended Generation | Swiss-Prot | Specific property or function | 2,529,006 | 11,911 |
| | RCSB PDB | Protein caption | 143,254 | 2,500 |
| Closed-set Answer | RCSB PDB | Specific property or function | 1,264,286 | 22,500 |
| | GO-BP | Biological process | 622,935 | 24,162 |
| | GO-MF | Molecular function | 404,629 | 5,891 |
| | GO-CC | Cellular component | 334,032 | 6,735 |
| | EC | Enzymatic catalytic activity | 176,942 | 2,278 |
| # All Instructions | | | 5,475,084 | 75,977 |
| # Supplemental Instructions (from TrEMBL) | | | + 5,253,440 | N/A |

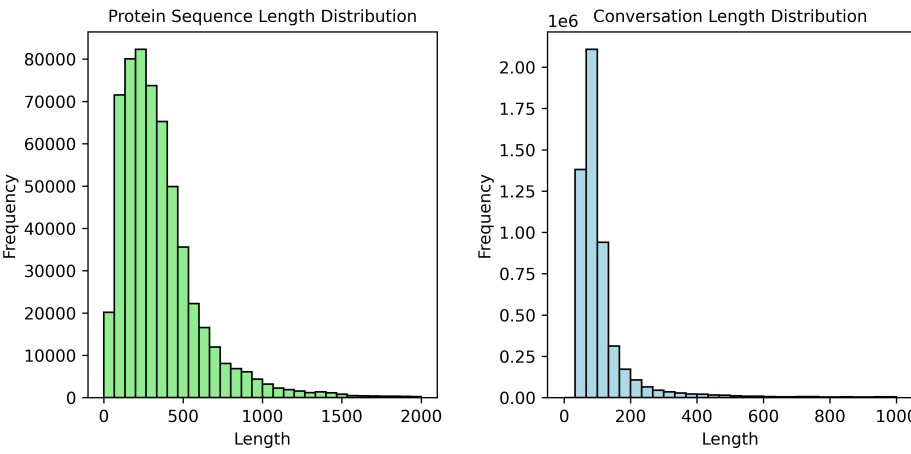

Figure 4: Distribution of protein sequence length and conversation length in the protein instruction dataset (training set).

The instruction template and example responses for Swiss-Port are as follows:

- **Function**

  - What is the primary function of <protein>?
  - What is the main function of <protein>?
  - What is the function of <protein>?
  - Explain the function of <protein>.
  - What is the characteristic function associated with the protein <protein>?
  - Can you define the function profile of the <protein>?
  - Give me the function caption of <protein>.

    **Example Response** Binds to muscle nicotinic acetylcholine receptor (nAChR) and inhibit acetylcholine from binding to the receptor, thereby impairing neuromuscular transmission. Produces peripheral paralysis by blocking neuromuscular transmission at the postsynaptic site. Has a lower toxicity than cobrotoxin.

- **Similarity**

  - Which protein family does <protein> belong to?
  - What is the protein family of <protein>?
  - What is the closest related protein family for <protein>?
  - Can you identify the family or group that <protein> belongs to?
  - To which protein family <protein> is classified?
  - Which protein class does <protein> fall into?

    **Example Response** Belongs to the snake three-finger toxin family. Short-chain sub-family. Aminergic toxin sub-subfamily.

- **Subcellular location**

  - Where is <protein> located in the cell?
  - Can you specify the subcellular location of <protein>?
  - What is the subcellular location of <protein>?
  - Could you describe the subcellular location of <protein>?
  - What are the primary subcellular regions where <protein> is detected?

    **Example Response** Colocalizes with ENA/VASP proteins at lamellipodia tips and focal adhesions, and F-actin at the leading edge. At the membrane surface, associates, via the PH domain, preferentially with the inositol phosphates, PtdIns(5)P and PtdIns(3)P. This binding appears to be necessary for the efficient interaction of the RA domain to Ras-GTPases (By similarity).

- **Induction**

  - Description the effects of environmental factors of <protein>'s expression.
  - What are the environmental factors that induce the expression of <protein>?
  - What environmental factors causes the upregulation of <protein>?
  - What are the environmental factors that lead to the upregulation of <protein>?

    **Example Response** Is slightly up-regulated when the bacterium is grown on t4LHyp or t3LHyp as sole carbon source.

- **Gene Ontology(Molecular Function)**

  - Which GO molecular function terms have <protein> been assigned to?
  - What molecular function is associated with <protein>?
  - Which GO terms outline the functional capabilities of <protein>?
  - What are the molecular functions of <protein>?

    **Example Response** ATP binding; protein serine kinase activity; protein serine/threonine kinase activity

- **Gene Ontology(Biological Process)**

  - Which GO biological process terms have <protein> been assigned to?
  - What biological process is associated with <protein>?
  - Which GO terms outline the biological processes of <protein>?

- What biological processes is <protein> involved in, based on gene ontology annotations?
- What are the biological processes of <protein>?

  **Example Response**  organic acid transmembrane transport; suberin biosynthetic process

- **Gene Ontology(Cellular Component)**

  - Which GO cellular component terms have <protein> been assigned to?
  - What cellular component is associated with <protein>?
  - Which GO terms outline the cellular components of <protein>?
  - What cellular components is <protein> involved in, based on gene ontology annotations?
  - What are the cellular components of <protein>?

    **Example Response**  cytosol; plant-type vacuole; plasma membrane

- **Developmental Stage**

  - At which specific developmental stages is <protein> expressed?
  - What are the developmental stages where <protein> is expressed?
  - What are the developmental stages where <protein> is detected?
  - What are the developmental stages where <protein> is found?

    **Example Response**  Detected at high levels at the tube tip during early pollen germination. In germinated pollen tubes it is localized in a punctate pattern throughout the cytoplasm but most prominently at the tip region.

- **Short Sequence Motif**

  - Can you identify and list all the motifs that are predicted to be present in <protein>?
  - What are the short sequence motifs that are predicted to be present in <protein>?
  - What are the short sequence motifs that are present in <protein>?
  - What are the short sequence motifs that are found in <protein>?

    **Example Response**  Nucleotide carrier signature motif

- **Tissue Specificity**

  - In which tissues is the expression of <protein> absent?
  - Describe the tissue-specific expression pattern of <protein>?
  - What is the tissue-specific expression pattern of <protein>?
  - What are the tissues where <protein> is expressed?

    **Example Response**  Expressed in the ciliated cells of the airway epithelium. Not detected in the mucous cells.

- **Activity Regulation**

  - Describe the activity regulatory mechanism of <protein> associated enzymes, transporters, microbial transcription factors.
  - What is the activity regulatory mechanism of <protein>?
  - Tell me about the activity regulatory mechanism of <protein>.

    **Example Response**  Activity is sensitive to salt concentration, a high concentration of KCL (500 mM) is needed for complete inactivation.

- **Pathway**

  - What is the role of <protein> in the metabolic pathway?
  - Which metabolic pathway does <protein> associate with?
  - What is the metabolic pathway that <protein> is involved in?

    **Example Response**  Ketone degradation; acetoin degradation.

The instruction template and example responses for RCSB PDB are as follows, where {GO} and {EC} are replaced with their actual meanings (text) corresponding to GO and EC annotations.

- **Caption**

- Tell me about this protein <protein>.
- Give me some information about <protein>.
- Give me the abstract of <protein>.
- Give me a comprehensive description of <protein>.
- Tell me about <protein>.

  **Example Response** The FANCM/FAAP24 heterodimer has distinct functions in protecting cells from complex DNA lesions such as interstrand crosslinks. These functions rely on the biochemical activity of FANCM/FAAP24 to recognize and bind to damaged DNA or stalled replication forks...

- **Others**
  - Does this protein contain non-polymer entities, <protein>?
  - Does this protein contain polymer entities, <protein>?
  - Does this protein contain DNA polymer entities, <protein>?
  - Does this protein contain RNA polymer entities, <protein>?
  - Does this protein contain solvent entities, <protein>?
  - Does this protein contain branched entities, <protein>?
  - Does this protein have unmodeled polymer monomers, <protein>?
  - Does this protein have hybrid nucleic acid polymer entities, <protein>?
  - Does this protein have cis-peptide linkages, <protein>?

    **Example Response** Yes./No.

The instruction template and example responses for other closed-set answer tasks are as follows:

- **EC**
  - Does <protein> associate with enzyme classification "{EC}"?
  - Does EC term "{EC}" outline the enzyme classifications of <protein>?
  - Is <protein> involved in enzyme classification "{EC}"?

    **Example Response** Yes./No.

- **GO-BP**
  - Does <protein> associate with biological process "{GO}"?
  - Does GO term "{GO}" outline the biological processes of <protein>?
  - Is <protein> involved in biological process "GO"?

    **Example Response** Yes./No.

- **GO-CC**
  - Does <protein> associate with cellular component "{GO}"?
  - Does GO term "{GO}" outline the cellular components of <protein>?
  - Is <protein> involved in cellular component "{GO}"?

    **Example Response** Yes./No.

- **GO-MF**
  - Does <protein> associate with molecular function "{GO}"?
  - Does GO term "{GO}" outline the functional capabilities of <protein>?
  - Does <pro tein> have molecular function "{GO}"?

    **Example Response** Yes./No.

## C SUPPLEMENT TO THE EXPERIMENTS

### C.1 IMPLEMENTATION DETAILS

As depicted in Figure 2, we extensively utilize group learning rates throughout the entire training pipeline of SEPIT. We tend to assign higher learning rates to randomly initialized parameters, while opting for lower learning rates for pre-trained parameters in order to mitigate forgetting, setting the ratio between lower and higher learning rates at 0.1. In Stage 0, we actually employ ESM2-650M (Lin et al., 2023) as the pLM and PubMedBert (Gu et al., 2021) as the text encoder to better encode biomedical text. We set the number of Gaussian Basis Kernel to 128. In Stage 1, we choose the representation from the penultimate layer of the protein encoder as input to LLMs to minimize the discrepancy between pre-training tasks and the current task. For the LLMs, we opt for TinyLlama-1.1B (Zhang et al., 2024). In Stage 2, we continue most of the settings from Stage 1, while setting the number of experts to 4, with Top-1 expert being activated at a time. At this stage, our protein encoder is frozen. Regarding the hyper-parameter settings for SEPIT-TinyLlama-MoEs, we set higher learning rate to $5e^{-5}$ and trained for 5 epochs in Stage 0. In Stage 1, we set higher learning rate to $2e^{-5}$ and trained for 1 epoch. In Stage 2, we set higher learning rate to $5e^{-5}$ and trained for 1 epoch. For all stages, we trained on 32 Tesla V100 GPUs, with a batch size per GPU set to 4, employing a warm-up and linear decay learning rate scheduler, and set the warm-up ratio to 0.06. In all experiments, we employe AMP, Zero Optimizer (Rasley et al., 2020) based on AdamW (Loshchilov & Hutter, 2019) and gradient checkpointing. For the hyper-parameter settings of other models, see Table 9. For all MoE models, we implement them based on DeepSpeed-MoE, employing expert parallelism during the training process and setting the expert parallel size to 4.

For the API-based models, we have tallied the number of tokens consumed in the experiments conducted for this paper. It is worth noting that due to the high cost associated with GPT-4 API requests, we randomly sampled 5% of the examples from the test set for testing. The specific token consumption is shown in Table 8. For all other models, we present the training hyper-parameter settings in Table 9, their training costs in Table 10, and their inference costs in Table 11.

Table 8: Tokens consumed by API-based models.

| API Model | Token Consumption |
|---|---|
| GPT-3.5-turbo | ∼19M |
| Claude-3-haiku | ∼19M |
| GPT-4-turbo | ∼0.95M * |

Table 9: Hyper-parameters of all models.

| Trained Model | Epochs | Wram-up Ratio | Batch Size per GPU | Global Batch Size | (Higher) Learning Rate | Auxiliary Loss Coefficient | Optimizer Stage |
|---|---|---|---|---|---|---|---|
| TinyLlama-Chat | 1 | 0.06 | 8 | 256 | $2e^{-5}$ | N/A | Zero 1 |
| OpenLlama-v2 | 1 | 0.06 | 8 | 512 | $4e^{-5}$ | N/A | Zero 3 |
| Llama-Chat | 1 | 0.06 | 6 | 192 | $2e^{-5}$ | N/A | Zero 3 |
| PIT-Stage 0 | 5 | 0.06 | 4 | 128 | $2e^{-5}$ | N/A | Zero 3 |
| PIT-TinyLlama-Stage 1 | 1 | 0.06 | 4 | 128 | $5e^{-5}$ | N/A | Zero 2 |
| PIT-TinyLlama-Stage 2 | 1 | 0.06 | 4 | 128 | $2e^{-5}$ | N/A | Zero 2 |
| PIT-TinyLlama-MoEs-Stage 2 | 1 | 0.06 | 4 | 256 | $1e^{-4}$ | 0.01 | Zero 2 |
| SEPIT-Stage 0 | 5 | 0.06 | 4 | 128 | $2e^{-5}$ | N/A | Zero 3 |
| SEPIT-Llama-Stage 1 | 1 | 0.06 | 2 | 128 | $5e^{-5}$ | N/A | Zero 3 |
| SEPIT-TinyLlama-Stage 1 | 1 | 0.06 | 4 | 128 | $5e^{-5}$ | N/A | Zero 2 |
| SEPIT-TinyLlama-Stage 2 | 1 | 0.06 | 4 | 128 | $2e^{-5}$ | N/A | Zero 2 |
| SEPIT-Llama-Stage 2 | 1 | 0.06 | 2 | 128 | $2e^{-5}$ | N/A | Zero 3 |
| SEPIT-TinyLlama-MoEs-Stage 2 | 1 | 0.06 | 4 | 128 | $5e^{-5}$ | 0.01 | Zero 2 |

Table 10: Training cost of all models.

| Trained Model | Parameter Size | Trainable Parameters | GPUs Cost (Hrs. $\times$ # V100) |
|---|---|---|---|
| TinyLlama-Chat | 1.1B | 1.1B | $44 \times 32$ |
| OpenLlama-v2 | 3B | 3B | $45 \times 64$ |
| Llama-Chat | 7B | 7B | $170 \times 32$ |
| PIT-Stage 0 | 650M + 110M | 650M | $20 \times 32$ |
| PIT-TinyLlama-Stage 1 | 1.1B + 650M | 1.1B + 650M | $20 \times 32$ |
| PIT-TinyLlama-Stage 2 | 1.1B + 650M | 1.1B | $50 \times 32$ |
| PIT-TinyLlama-MoEs-Stage 2 | 3.2B + 650M | 3.2B | $68 \times 64$ |
| SEPIT-Stage 0 | 650M + 110M | 650M | $26 \times 32$ |
| SEPIT-Llama-Stage 1 | 7B + 650M | 7B | $82 \times 64$ |
| SEPIT-TinyLlama-Stage 1 | 1.1B + 650M | 1.1B | $30 \times 32$ |
| SEPIT-TinyLlama-Stage 2 | 1.1B + 650M | 1.1B | $50 \times 32$ |
| SEPIT-Llama-Stage 2 | 7B + 650M | 7B | $220 \times 64$ |
| SEPIT-TinyLlama-MoEs-Stage 2 | 3.2B + 650M | 3.2B | $126 \times 32$ |

Table 11: Inference cost of all models.

| Inferenced Model | Parameter Size | Activated Parameters | GPUs Cost (Hrs. $\times$ # T4) |
|---|---|---|---|
| Galactica-base | 1.3B | 1.3B | $2 \times 8$ |
| BioMedGPT | 7B | 7B | $11 \times 8$ |
| TinyLlama-Chat | 1.1B | 1.1B | $1 \times 8$ |
| OpenLlama-v2 | 3B | 3B | $8 \times 8$ |
| Llama-Chat | 7B | 7B | $11 \times 8$ |
| PIT-TinyLlama | 1.1B + 650M | 1.1B + 650M | $1.25 \times 8$ |
| PIT-TinyLlama-MoEs | 3.2B + 650M | 1.1B + 650M | $1.5 \times 8$ |
| SEPIT-TinyLlama | 1.1B + 650M | 1.1B + 650M | $1.5 \times 8$ |
| SEPIT-Llama | 7B + 650M | 7B + 650M | $21 \times 8$ |
| SEPIT-TinyLlama-MoEs | 3.2B + 650M | 1.1B + 650M | $1.75 \times 8$ |

## C.2 METRIC EXPLANATION

**BLEU score:** The BLEU (Bilingual Evaluation Understudy) score (Sutskever et al., 2014) is a metric used to evaluate the quality of machine-translated text against human-translated reference texts, which is calculated using n-gram precision. The general formula for calculating BLEU score is as follows, where **BP** penalize overly short translations:

$$p_n = \frac{\sum_{C \in \{ \text{Candidates} \}} \sum_{\text{n-gram} \in C} \text{Count}_{\text{clip}} ( \text{n-gram} )}{\sum_{C' \in \{ \text{Candidates} \}} \sum_{\text{n-gram}' \in C'} \text{Count} ( \text{n-gram}' )}, \tag{16}$$

$$\mathbf{BP} = \begin{cases} 1 & \text{if } c > r \\ e^{(1-r/c)} & \text{if } c \leq r \end{cases}, \tag{17}$$

$$\textbf{BLEU} = \textbf{BP} \cdot \exp\left(\sum_{n=1}^{N} w_n \log p_n\right), \tag{18}$$

$$\log \textbf{BLEU} = \min(1 - \frac{r}{c}, 0) + \sum_{n=1}^{N} w_n \log p_n. \tag{19}$$

**ROUGE-N score:** ROUGE-N (Lin, 2004) is a widely-used automatic text evaluation metric designed to compare the similarity between generated text and reference text, which can be considered an improved version of BLEU with a focus on recall rather than precision. The general formula for calculating ROUGE score is as follows:

$$\textbf{ROUGE} - \textbf{N} = \frac{\sum_{C \in \{\text{ Candidates }\}} \sum_{\text{n-gram } \in C} \text{Count}_{\text{match}}(\text{ n-gram })}{\sum_{C' \in \{\text{ Candidates }\}} \sum_{\text{n-gram }' \in C'} \text{Count}(\text{ n-gram }')}. \tag{20}$$

**ROUGE-L score:** ROUGE-L (Lin, 2004) computes the overlap of the longest common subsequence (LCS) between the produced text and the standard references as follows, where $X$ represents the standard answer, and $Y$ denotes the generated answer, with their respective lengths being $n$ and $m$. $\beta$ is a hyper-parameter used to adjust the focus between precision $P_{\text{lcs}}$ and recall $R_{\text{lcs}}$:

$$R_{\text{lcs}} = \frac{LCS(X, Y)}{m}, \tag{21}$$

$$P_{\text{lcs}} = \frac{LCS(X, Y)}{n}, \tag{22}$$

$$\textbf{ROUGE} - \textbf{L} = \frac{(1 + \beta^2) R_{lcs} P_{lcs}}{R_{lcs} + \beta^2 P_{lcs}}. \tag{23}$$

**METEOR score:** METEOR (Banerjee & Lavie, 2005) addresses certain inherent shortcomings of the BLEU score by taking into account both precision and recall evaluated over the entire corpus. The general formula for calculating METEOR score is as follows, where Penalty is the penalty of excessive word mismatches and $\alpha$ is a hyper-parameter:

$$F = \frac{(\alpha^2 + 1)P}{R + \alpha P}, \tag{24}$$

$$\textbf{METEOR} = (1 - \text{Penalty}) \cdot F. \tag{25}$$

**BERT score:** BERT score (Zhang et al., 2019) is an automatic evaluation metric for text generation, which computes a similarity score for each token in the candidate sentence with each token in the reference sentence. The general formula for calculating BERT score is as follows, where tokens of reference sentence $x$ and candidate sentence $\hat{x}$ are represented by contextual embeddings:

$$R_{\text{BERT}} = \frac{1}{|x|} \sum_{x_i \in x} \max_{\hat{x}_j \in \hat{x}} \mathbf{x}_i^\top \hat{\mathbf{x}}_j, \tag{26}$$

$$P_{\text{BERT}} = \frac{1}{|\hat{x}|} \sum_{\hat{x}_j \in \hat{x}} \max_{x_i \in x} \mathbf{x}_i^\top \hat{\mathbf{x}}_j, \tag{27}$$

$$F_{\text{BERT}} = 2 \frac{P_{\text{BERT}} \cdot R_{\text{BERT}}}{P_{\text{BERT}} + R_{\text{BERT}}}. \tag{28}$$

### C.3 MORE PERFORMANCE COMPARISON

Considering that there are additional related works capable of captioning input protein sequences but lacking instruction-following capabilities, we attempted to compare performance on overlapping properties and functions with these works. Table 12 shows a performance comparison with ProtT3 (Liu et al., 2024b), which is the most representative of these works. For open-ended generation and closed-set answer tasks, we selected Function, Similarity, Subcellular location, and Q&A related to structure and properties for comparison. The results demonstrate that our model exhibits significantly better performance.

Table 12: Performance comparisons on overlapping properties and functions.

| Model | Activated Parameters | Open-ended | | | | | | | | | Closed-set |
|---|---|---|---|---|---|---|---|---|---|---|---|
| | | BLEU-2 | BLEU-4 | ROUGE-1 | ROUGE-2 | ROUGE-L | METEOR | BERT-P | BERT-R | BERT-F1 | Accuracy |
| ProtT3 | 1.3B | 65.38 | 51.87 | 73.75 | 57.88 | 72.88 | 70.39 | 96.29 | 95.80 | 96.03 | 90.52% |
| **SEPIT-TinyLlama-MoEs** | 1.8B | **83.07** | **81.29** | **86.74** | **83.55** | **86.05** | **85.96** | **98.04** | **98.01** | **98.02** | **94.92%** |

### C.4 MORE ABLATION STUDIES

For the data ablation experiments, we also tested different forms of protein inputs, with the results shown in Table 13. For both forms of protein inputs, the addition of extra low-quality data does not result in performance improvement but instead led to performance degradation.

Table 13: Data ablation study on SEPIT's pre-train data.

| Dataset | Infer w/ Struct. | Open-ended Generation | | | | Closed-set Answer |
|---|---|---|---|---|---|---|
| | | BLEU-2 | ROUGE-L | METEOR | BERT-F1 | Accuracy |
| Protein Instrcution Dataset (5.47M) | ✓ | 60.28 | 71.13 | 68.27 | 95.64 | 79.73% |
| | ✗ | 59.98 | 70.87 | 68.00 | 95.59 | 79.53% |
| w/ TrEMBL (+5.25M) | ✓ | 58.71 | 69.65 | 66.93 | 95.42 | 79.52% |
| | ✗ | 58.55 | 69.54 | 66.78 | 95.40 | 79.48% |

### C.5 MORE CASE STUDIES

**More visualization for workload of experts in SEPIT.** More visualization for workload of experts in SEPIT is shown in Figure 5 and 6. The results are similar to our analysis in the main text.

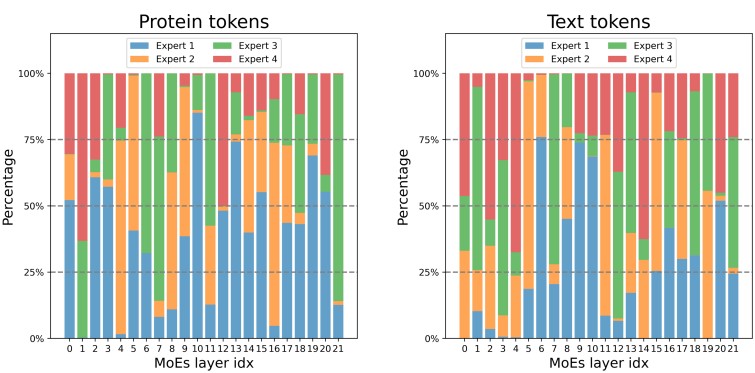

Figure 5: Workload of experts in SEPIT for protein tokens and text tokens.

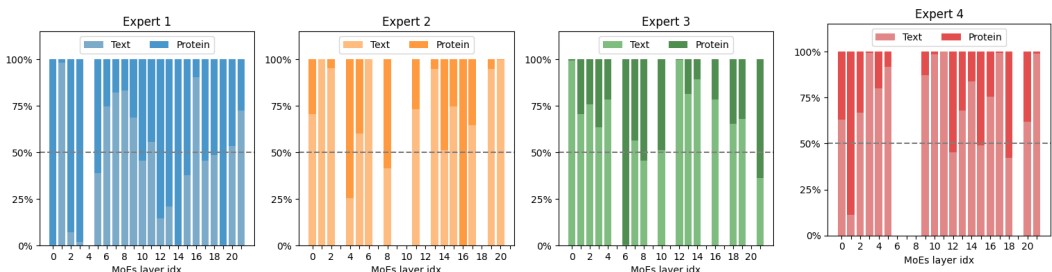

Figure 6: Workload of each expert in SEPIT for protein tokens and text tokens.

**More cases for general-purpose protein understanding ability of SEPIT.** More cases of comparison is shown in Table 14.

Table 14: More cases on general-purpose protein understanding ability of SEPIT.

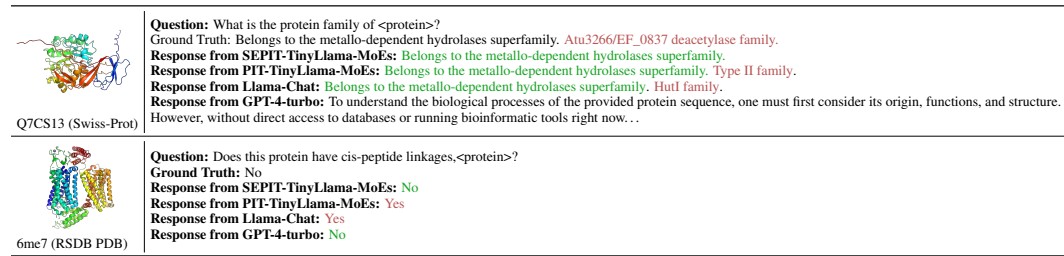

More cases of our proposed SEPIT-TinyLlama-MoEs:

- **O43143 (Swiss-Prot)**
  - **Instruction:** What are the short sequence motifs that are present in <protein>?
  - **Ground Truth:** DEAH box
  - **Response:** DEAH box

- **Q8DPM9 (Swiss-Prot)**
  - **Instruction:** What is the function of <protein>?
  - **Ground Truth:** Bidirectionally degrades single-stranded DNA into large acid-insoluble oligonucleotides, which are then degraded further into small acid-soluble oligonucleotides.
  - **Response:** Bidirectionally degrades single-stranded DNA into large acid-insoluble oligonucleotides, which are then degraded further into small acid-soluble oligonucleotides.

- **B4LM02 (Swiss-Prot)**
  - **Instruction:** Give me the function caption of <protein>.
  - **Ground Truth:** Plays a central role in 2-thiolation of mcm(5)S(2)U at tRNA wobble positions of tRNA(Lys), tRNA(Glu) and tRNA(Gln). Directly binds tRNAs and probably acts by catalyzing adenylation of tRNAs, an intermediate required for 2-thiolation. It is unclear whether it acts as a sulfurtransferase that transfers sulfur from thiocarboxylated URM1 onto the uridine of tRNAs at wobble position.
  - **Response:** Plays a central role in 2-thiolation of mcm(5)S(2)U at tRNA wobble positions of tRNA(Lys), tRNA(Glu) and tRNA(Gln). May act by forming a heterodimer with ctu1/atpbd3 that ligates sulfur from thiocarboxylated urm1 onto the uridine of tRNAs at wobble position.

- **Q98QW4 (Swiss-Prot)**
  - **Instruction:** Which protein family does <protein> belong to?

- **Ground Truth:** In the N-terminal section; belongs to the phosphoglycerate kinase family.
- **Response:** Belongs to the phosphoglycerate kinase family.

- **P0DMD4 (Swiss-Prot)**

    - **Instruction:** Which protein family does <protein> belong to?
    - **Ground Truth:** Belongs to the scolopendra neurotoxin 10 family.
    - **Response:** Belongs to the scolopendra neurotoxin 02 (Dtx-II) family.

- **A1VKS2 (Swiss-Prot)**

    - **Instruction:** What is the closest related protein family for <protein>?
    - **Ground Truth:** Belongs to the pantothenate synthetase family.
    - **Response:** Belongs to the pantothenate synthetase family.

- **P67911 (Swiss-Prot)**

    - **Instruction:** What molecular function is associated with <protein>?
    - **Ground Truth:** ADP-glyceromanno-heptose 6-epimerase activity; NADP binding
    - **Response:** ADP-glyceromanno-heptose 6-epimerase activity; NADP binding

- **Q9X5Y0 (Swiss-Prot)**

    - **Instruction:** What cellular component is associated with <protein>?
    - **Ground Truth:** bacterial-type flagellum basal body
    - **Response:** bacterial-type flagellum basal body

