# OpenReview forum: "Structure-Enhanced Protein Instruction Tuning: Towards General-Purpose Protein Understanding"
_ICLR.cc/2025/Conference — Submitted to ICLR 2025_

### Official Review · Reviewer_yPkE · 2024-11-03

**Soundness:** 2
**Presentation:** 3
**Contribution:** 2
**Rating:** 5
**Confidence:** 4

**Summary:**

The excerpt describes a framework called Structure-Enhanced Protein Instruction Tuning (SEPIT) aimed at enabling large language models (LLMs) to achieve general-purpose protein understanding. The authors argue that existing protein language models (pLMs), even when fine-tuned, are often task-specific and struggle to provide a holistic understanding of protein properties and functions. By integrating structural information, employing a novel instruction tuning pipeline, and utilizing a comprehensive dataset,

**Strengths:**

* i) Incorporates Structural Information: this work includes a structure-aware module that lets it use both sequence data (which is plentiful) and structural data (which is scarcer) to understand proteins. the idea structural information could help understanding is ituiative and sounded.

* ii) Positive Performance on Function Prediction Tasks：Experimental results show that SEPIT performs better than other methods on tasks involving both open-ended questions (like explaining the function of a protein) and closed-set questions (like determining if a protein has a specific function)

**Weaknesses:**

i) Data Leakage Concerns:
* Considering the redundancy and similarity of protein data, a large amount of work indicates the importance of data cleaning. I believe the current experiment fails to address my concerns about data leakage. For instance, AlphaFold2, AlphaFold3, and Prot2Text propose methods to remove 40% sequence similarity. Such an approach would likely result in lower (and more reasonable) BLEU/ROUGE scores. It is recommended that the authors further refine the data splitting to mitigate potential data leakage issues.

ii)  In the current experimental design, many of the studies mentioned by the authors were not included, such as Prot2Text and ProteinChat. Moreover, including some updated data (updated validation sets) could further enhance the model's credibility. The argue that these methods are limited to specific tasks like prediction and retrieval and don't offer the open-ended generation capabilities required for comprehensive protein understanding. However, a quantative comparision would be appriciate.

**Questions:**

i) The necessity and advantages of the proposed two-stage training process require further clarification. What are the potential drawbacks of a single-stage approach? How does the two-stage process specifically address these limitations? What empirical evidence supports this design choice?"

ii) While the evaluation metrics derived from LLM contexts provide valuable insights, the paper would benefit from a more comprehensive analysis of model performance using domain-specific evaluation criteria. I suggest author could considering :
Stratifying the evaluation set based on biological classification hierarchies and analyzing performance patterns across different biological categories

---

### Official Review · Reviewer_K4CN · 2024-11-03

**Soundness:** 4
**Presentation:** 3
**Contribution:** 3
**Rating:** 6
**Confidence:** 4

**Summary:**

This is an excellent work! The authors propose a novel instruction tuning method to understand the protein structure and connect the PLM with LLM to provide the natural language explanation and human interaction. Meanwhile, a very large scale dataset is curated by them which is also a very significant contribution.

**Strengths:**

1. The method is very novel and bridges the gap in this domain.
The motivation for this work is very strong, which bridges the gap between protein and language. They also propose a two-stage method to achieve this.
2. Large-scale dataset is curated.
High-quality, large-scale, and AI-ready data are always needed in this field.
3. Extensive experiments and competitive performance.
From Tab 1, we can find the performance is very competitive compared to SOTA models.

**Weaknesses:**

1. Maybe add more case studies, (only 2 seems too few).
My suggestion is to categorize the question type (currently only two types of questions are made). If the author can summarize and categorize most questions and do the case studies for each of them. That would be great.
2. The in-depth analysis is not enough. Add more analysis in the experiments part.
There is a lack of error analysis. Can you also add error analysis in the case studies? So that we know when the model might make mistakes.

**Questions:**

See details in weakness.

---

### Official Review · Reviewer_jsaQ · 2024-11-05

**Soundness:** 3
**Presentation:** 3
**Contribution:** 3
**Rating:** 6
**Confidence:** 3

**Summary:**

This paper presents a novel method for general protein understanding, termed SEPIT. The approach integrates a structure-aware module into protein language models (pLMs) and subsequently connects these structure-enriched pLMs to large language models (LLMs) to enhance the understanding of proteins. Building on this model framework, the authors propose a two-stage instruction tuning process. Initially, a foundational understanding of proteins is established through title-based instructions, which is then refined using a mixture of experts (MoEs) to capture more complex attributes and functional information.The authors also constructed a comprehensive dataset for both open-ended generation and closed-ended answering, based on Swiss-Prot and RCSB PDB. Extensive experimental results demonstrate that SEPIT significantly outperforms state-of-the-art models, and ablation studies provide structural validation of the effectiveness of each component within SEPIT.

**Strengths:**

1.I appreciate the clear and well-structured presentation of the paper. Their writing style is fluid, effectively conveying the motivations behind the research and the logical progression of the methodology.
2.The experiments are comprehensive, featuring insightful ablation studies and case studies. The authors articulate the effectiveness of various techniques clearly, ensuring that these methods remain accessible and not overly complex.
3.The authors have successfully achieved the objectives outlined in the paper, and the figures are exceptionally clear. I appreciate their thoughtful summary of their contributions.

**Weaknesses:**

I mainly have three concerns:
1.Although the structural components serve their purpose, is such a design too simplistic to capture more detailed structural information? As is well known, the structure of proteins is highly complex, containing numerous amino acids. Could this approach become unreliable and introduce some errors?
2.The experimental results do not provide sufficient evidence to ascertain whether the mixture of experts (MoEs) module is functioning as intended. Further clarity on its performance would strengthen the findings.
3.What are the innovative aspects of the newly constructed dataset compared to the previous datasets? Are there any other models that have utilized this new dataset?

**Questions:**

1.What are the inference time and complexity of this framework?
2.Is the model capable of explaining interactions between two proteins or other more complex scenarios?
3.Is the dataset constructed by the authors representative in this field?
4.Not a question, What does the author suggest for addressing the resource consumption caused by excessively long protein sequences?

---

### Official Review · Reviewer_UgeA · 2024-11-09

**Soundness:** 3
**Presentation:** 4
**Contribution:** 3
**Rating:** 8
**Confidence:** 4

**Summary:**

The paper introduces structure-enhanced protein instruction tuning (SEPIT) framework to learn a general-purpose protein understanding model. The paper combines a structure-aware protein encoder with a pretrained large language model, then trains a mixture of experts. To train the new model, they create a new large-scale protein instruction tuning dataset from existing resources. The experiments on the test split of their dataset shows that the structure aware training improves model performance. The paper includes comprehension ablation to show the importance of each component in the model architecture. Finally, they include qualitative analysis regarding how the mixture of experts is utilized at inference time.

**Strengths:**

- The paper is very well written.
- The architecture is well-motivated. The paper clearly explains each component, making it easy to understand and reproduce.
- The experiment section is comprehensive and up to date. The paper compares SEPIT with recent large language models and PIT models. In all cases, they show that SEPIT achieves the highest performance.
- The paper includes key ablations that test the robustness of the SEPIT model. For example, Table 4 shows that SEPIT does not significantly drop performance even when 3D structure information is unavailable.

**Weaknesses:**

**Generalization beyond the training data.**
The paper does not experiment on datasets other than its dataset. In Table 1, the paper includes results on the test set of its newly constructed dataset. This experiment does not test the generalization of the model to new datasets, a key practical requirement. It would increase the impact of the paper if experiments with more datasets were included. This is the main weakness of the paper.

**A need for warm up stage.**
The paper includes a warm up for the protein encoder in Stage 0. In this stage, the paper pretrains the encoder with a self supervised learning objective, i.e., denoising objective. However, it is unclear from the experiments if this stage is necessary. The motivation of the pretrained encoder will be randomly initialized is not a strong justification for this additional step. Can the protein encoder be jointly trained with the language model?

**Combination of existing components.**
This is a very minor weakness. The paper combines components from existing literature to create a hybrid language model. The architectural innovation is limited. This is not a deal breaker, but it would be great if any existing modules could be simplified.

Typos
- Line 126: Incorrect citation for ALBEF
- Line 442: Remove full stop after GO annotation tasks

**Questions:**

Please see the weaknesses in the paper.

Additional questions:
- Nit: In equation 11, what is $W_{m}$?
- How are the instructions provided to the zero-shot models?

---

### Meta-Review · Area_Chair_UyMJ · 2024-12-23

**Metareview:**

The paper introduces the Structure-Enhanced Protein Instruction Tuning (SEPIT) framework, which aims to achieve general-purpose protein understanding by integrating structural knowledge into protein language models and connecting them to large language models. The authors propose a two-stage instruction tuning pipeline and construct a novel comprehensive protein instruction dataset. Experimental results demonstrate that SEPIT outperforms existing models in both open-ended generation and closed-set answer tasks.

Strengths:
* **Novel, Compelling Framework**. The proposed SEPIT framework effectively combines structural information (scarce) with sequence protein data (plentiful), enhancing protein understanding.
* **Comprehensive Experimental Validation**. Extensive experiments show that the proposed method works well across various task.
* **Clear Presentation**. The paper is well-written and clearly explains the methodology and experimental results.
* **A novel dataset**. The proposed protein instruction dataset containing open-ended generation and a closed-set answer task might be of independent interest.

Weaknesses:
* **Unclear Generalization**. The paper focuses most experiments on its own dataset. Only during the rebuttal phase were experiments on external out-of-distribution datasets reported, but these were reported with baseline comparison and not included in the revised version, despite the request from reviewer UgeA . This leaves the question of generalization mostly unanswered.
* **Complexity of Structural Modeling**. As pointed out by jsaQ, the structural components might be too simplistic to capture detailed structural information, potentially introducing errors.
* **Two-Stage Training Justification**: The proposed two-stage training process is not sufficiently motivated and justified.
* **Limited / no error analysis**

Reason to reject: The lack of comprehensive experiments on external datasets, which renders the performance of the method in practice uncertain.

**Additional Comments On Reviewer Discussion:**

There was limited engagement from the reviewers, despite repeated pleas from the authors. The main points raised in the reviews/rebuttal were:

* Data Leakage Concerns: Reviewer yPkE raised concerns about potential data leakages due to redundancy and similarity in protein data. The authors clarified that they followed standard practices to avoid data leakage by ennsuring there are no identical proteins between training/test sets.
* Baseline Models: Reviewer yPkE noted the lack of comparisons with certain baselines. The authors responded by highlighting additional baseline comparisons in the appendix and provided results on updated test sets.
* Two-Stage Training Process: Reviewer yPkE questioned the necessity of the two-stage training process. The authors explained that this approach is common in MoE-related literature and provided additional justification for its use.
* Generalization Beyond Training Data: Reviewer UgeA and Reviewer jsaQ emphasized the need for experiments on external datasets. The authors added results on new test sets and out-of-distribution datasets to address this concern.
* Structural Modeling: Reviewer jsaQ and Reviewer K4CN raised concerns about the simplicity of the structural modeling approach. The authors acknowledged this and suggested future work to explore more advanced structural modeling methods.

For reviewers that did not engage with the authors' rebuttal and whose concerns appear to have been resolved, those concerns were downweighed.

---

### Decision · Program_Chairs · 2025-01-22

Reject